# Evaluating User Safety Aspects of AI-Based Systems in Industrial Occupational Safety: A Critical Review of Research Literature

**DOI:** 10.3390/ijerph22050705

**Published:** 2025-04-29

**Authors:** Jaroslava Huber, Bernhard Anzengruber-Tanase, Martin Schobesberger, Michael Haslgrübler, Robert Fischer-Schwarz, Alois Ferscha

**Affiliations:** 1Pro2Future GmbH, 4040 Linz, Austria; bernhard.anzengruber@pro2future.at (B.A.-T.); michael.haslgruebler@pro2future.at (M.H.); 2Institute of Pervasive Computing, Johannes Kepler University, 4040 Linz, Austria; martin.schobesberger@pervasive.jku.at (M.S.); ferscha@pervasive.jku.at (A.F.); 3Department of Machine Safety, AUVA, 1100 Vienna, Austria; robert.fischer-schwarz@auva.at

**Keywords:** artificial intelligence, occupational safety, safe AI, sensors, human-centered AI

## Abstract

AI technologies are becoming increasingly prevalent in industrial workplaces, extending their applications beyond productivity to critical areas such as occupational safety. From our perspective, it is important to consider the safety of these AI systems for users already at the research and development stage, rather than only after deployment. Therefore, in this review, we synthesize publications that propose such AI-based safety systems to assess how potential risks are addressed early in their design and prototype stages. Consequently, we explore current advancements in AI-driven, sensor-based, and human-centered applications designed to enhance occupational safety by monitoring compliance, detecting hazards in real time, or assisting users. These systems leverage wearables and environmental sensing to proactively identify risks, support decision-making, and contribute to creating safer work environments. In this paper, we categorize the technologies according to the sensors used and highlight which features are preventive, reactive, or post-incident. Furthermore, we address potential risks posed by these AI applications, as they may introduce new hazards for workers. Through a critical review of current research and existing regulations, we identify gaps and propose key considerations for the safe and ethical deployment of trustworthy AI systems. Our findings suggest that in AI- and sensor-based research applications for occupational safety, some features and risks are considered notably less than others, from which we deduce that, while AI is being increasingly utilized to improve occupational safety, there is a significant need to address regulatory and ethical challenges for its widespread and safe adoption in industrial domains.

## 1. Introduction

### 1.1. AI-Based Systems for Occupational Safety

Industry 4.0 goes beyond interconnectivity and digital manufacturing. It promises increases in productivity through the integration of digital production systems with the analysis and communication of all data generated within an intelligent environment [1]. Therefore, it has driven significant research and development in AI-based sensor technologies for the industrial sector. These advancements have proven effective in addressing the challenges manufacturers face today, since modern manufacturing environments are becoming more dynamic and connected, but also more complex due to increased interdependencies, uncertainties, and the vast amounts of data being generated [2].

AI-based deployments such as embedded systems, wearables, and Internet of Things (IoT) technologies, although often regarded as tools to increase productivity, also carry large potential for improving worker safety. There has been growing interest in integrating AI in the framework of the interdisciplinary field of occupational safety and health (OSH), which focuses on the prevention of workplace injuries and illnesses [3]. Various potentially beneficial features of emerging technologies in the context of Industry 4.0, including the creation of safer work environments, enhanced health monitoring, and the reduction in occupational hazards, are known and sought after. However, alongside these aspects, there are emerging challenges, such as elevated stress levels, fatigue, musculoskeletal disorders, and increased psychosocial risks [4]. The tools developed for the scope of reducing accidents at the workplace aim to detect, manage, and eliminate occupational hazards. Conceptually, but also in practice, this is commonly done by monitoring the human operator (alarming the person about the threat and providing contextual information, support, and assistance) or monitoring the occupational environment, or a combination of those [5]. One key example, currently the focus of extensive research, is computer vision technologies, which enhance safety management by improving hazard identification and workers’ situational awareness [6]. As the EU envisions Industry 5.0 as an improved, human-centered evolution of Industry 4.0, placing the human operator at the center of digitization will become a non-negotiable requirement [7]. Besides this, it will be necessary to develop technologies that enhance and support human performance, which makes the focus on occupational health and safety crucial in future production processes [8]. As the industrial adoption of advanced AI-based solutions remains relatively limited beyond experimental pilot stages, real-world environments present unique and complex challenges that many organizations are not yet prepared for [2]. In our research, we acknowledge the growing need to strengthen the relationship and collaboration between human operators and AI-based technology. One way to achieve this is by building trust in the technology and, more broadly, aim for the highest standards of workplace safety through the design of reliable systems. For the design of reliable and trustworthy systems, it is necessary to explore where potential hazards and risks of AI-based technologies may arise, so they can be prevented. Consequently, in the context of this work, we examine AI-based applications that assist or monitor the human operator and investigate them from the point of view of how the AI-based technologies themselves might pose hazards to the worker.

### 1.2. Related Research

Our review of the literature on AI systems for workplace safety in industrial settings shows that many studies focus more on coverage than detail. For example, Flor et al. [9] and Patel et al. [5] examine a wide range of scientific papers on AI technologies for safety across various industrial sectors. Patel et al., in particular, focus on wearable technologies and connected-worker solutions, highlighting the numerous potentials and opportunities of these emerging systems. However, while they acknowledge the difficulty in assessing the safety and efficacy of many workplace technologies currently on the market, their analysis does not prioritize a critical evaluation of these technologies’ challenges or limitations. Fisher et al. [10] conduct a scoping review on the impact of AI on occupational safety and health (OSH) inequities, pointing out discrepancies in how technological advancements are distributed across different social groups, industries, and geographical regions. Although they emphasize the need for future research on how AI-based systems affect the health and safety of operators, their review does not systematically address how these technologies themselves may introduce new risks. Akyıldız [11] provides an interesting analysis of the transformation of OSH practices in the digital age, focusing on both the advantages and disadvantages of AI components. However, this paper lacks a systematic foundation supported by established frameworks or widely accepted norms.

Following the extensive review by Pishgar et al. [3], which examined AI-based safety applications across five industrial sectors (oil and gas, mining, transportation, construction, and agriculture) and identified key research gaps, we chose to focus on recent developments, analyzing publications from 2022 and later. Pishgar et al.’s framework, which outlines how AI systems manage and control exposure risks in the workplace, particularly in terms of when do these systems intervene, was an important reference point that we adopted.

The primary contribution of this survey is in its approach to analyzing AI-based safety systems that are human-centered, either assisting or monitoring the operator. In addition to offering a structured categorization based on the type of sensor monitoring, our paper critically examines whether these systems potentially address the new challenges they may introduce, particularly in terms of defined safety risks. To our knowledge, no existing research has combined a technical analysis with an initial risk evaluation framework for AI-based safety systems in this way, making a step towards more comprehensive risk assessments in the field.

## 2. Materials and Methods

### 2.1. Publication Inclusion Criteria

In this study, we employ a systematic approach by gathering and analyzing research papers on applications of AI-based systems which are designed to enhance occupational safety. We focus on systems that are built around the human operator and follow a human-centered design, i.e., they are developed with an emphasis on user interaction and address human needs and experiences [12]. Specifically, we are interested in applications where the human operator is either (i) assisted by the system, where the system provides safety-relevant information (e.g., safety wearables worn on the body) or (ii) monitored by the system (e.g., computer vision technologies monitoring safety compliance at a shop floor). Generally, these systems are frequently composed of a similar set of components, which frequently include, but are not limited to, e.g., a power unit, a sensor unit, a controller unit, a connection unit, a storage unit, and a type of user interface. While these components might simply describe a computer, this is important to mention here, since we focus our discussion largely on the sensor unit. The research logic is demonstrated in Figure 1.

AI-based systems are increasingly being applied in domains beyond the scope of this research, such as predictive maintenance and robotics. In predictive maintenance, AI focuses on machine health rather than augmenting human abilities (which would be the human-centered approach), as it monitors equipment and predicts failures. Robotics, even when limited to human-centered applications, is a broad field that falls outside the focus of this study. Therefore, in selecting literature, we deliberately concentrated on embedded systems, IoT frameworks, and assistance and monitoring systems to maintain a focused and manageable scope.

Therefore, we only examined publications that met each of the following inclusion criteria:Research papers that explore the application of AI-driven sensor systems published in English language.The applications are built around the human operator and follow a human-centered design.Applications where the human operator is either assisted by the system or monitored by the system.Applications used in different occupational industrial environments.Articles published after 2022.

### 2.2. Publication Selection Process

Using Google Scholar, IEEE Xplore, and ACM Digital Library, we focused on papers published from 2022 onward, with the rationale detailed in the Related Research section. The keywords used included “occupational safety”, “artificial intelligence”, “wearable systems”, “smart personal protective equipment (PPE)”, “cognitive systems”, “safety recognition”, “industry”, and combinations of these terms. From the initial search, 155 abstracts were selected and reviewed, resulting in a full-text examination of 56 papers. Out of these, 20 publications were included in the study. Papers were excluded if they were duplicates, inaccessible, did not meet the inclusion criteria, or were not research articles (for details, see Figure 2).

To provide additional context and benchmark research findings, we also examined commercial applications of AI-based systems for occupational safety, assessing which technologies have gained traction in the market. For this, we selected ten applications based on the top results from an online search. While not entirely random, this method ensured a diverse and commonly referenced selection. These results were not fully incorporated into the study but were considered in part to address a specific aspect of one of the research questions.

### 2.3. Categorizing Publications and Formulating Research Questions

Considering their frequent occurrence in the literature, we organized the applications into two primary categories: (i) wearable systems and (ii) systems deployed in the environment (see Table 1). Consequently, the applications were grouped according to the type of sensor and data that were collected. In the category of wearable systems, the four groups physiological monitoring (e.g., vital parameters), environmental monitoring worn on the body (e.g., air quality, temperature), movement and posture monitoring (e.g., accelerometers), and proximity and location tracking (e.g., GPS) solidified. In the category of systems deployed in the environment, we established the following groups: image sensors (e.g., computer vision), environmental sensors deployed in the environment (e.g., noise, light, air quality), and motion and proximity sensors (e.g., infrared). With this, we created a solid organizational structure to categorize the analyzed applications.

Simultaneously, we categorized the applications according to their belonging into the “preventive”, “reactive”, or “post-incident” groups, these groups indicating at which moment the AI-based system intervenes or operates. This follows the approach of Pishgar et al. [3]. The reasoning behind this is as follows: AI-based systems have the potential to be designed in a holistic way, i.e., in the context of workplace injuries, they can contribute to the detection, prevention, and reaction [13]. We investigated which role AI-based applications play in the anticipation and control of the risks, categorizing them into AI-based technologies that either operate (i) preventively (keeping the worker in the safe state), (ii) reactively (react to a hazardous situation to bring the worker back into the safe state), or (iii) provide measures to mitigate the damage after a hazardous incident occurred. For this part specifically, we analyzed research applications and, additionally, some applications on the market, sorting them, likewise, according to their contribution on a preventive, reactive, or post-incident level. Furthermore, we looked into the technologies behind them and potential risks these technologies might exert on the worker. We formulated the following two research questions, which will be discussed in the next section:
RQ1: How can AI-based systems and technologies implemented for occupational safety in industrial settings be categorized based on the following aspects?RQ1.1: Which AI-based wearable systems and technologies are implemented for occupational safety in industrial settings?RQ1.2: Which AI-based systems and technologies deployed in the environment are implemented for occupational safety in industrial settings?RQ1.3: Which preventive, reactive, and post-incident features of these AI-based safety applications can be identified?RQ2: Given that some AI-based applications for occupational safety inherently introduce new risks while others do not, does the respective scientific literature address these risks? Additionally, when considering both explicitly addressed risks and those not posed by the design of the respective systems, which applications can be considered safer or less safe overall?

## 3. Results

The following section presents the key findings of this research. It outlines the results and insights gained from the analysis, providing a comprehensive overview of the main outcomes.

### 3.1. Table Summarizing Findings for RQ1

To address RQ1, we categorized the relevant papers into groups, as shown in Table 1. The three subcomponents of the question will be examined in the subsequent sections.

**Table 1 ijerph-22-00705-t001:** “Type of Sensor/Data Collected”, categorized by the “Stage of Intervention”. This table summarizes the findings for RQ1 and its three subparts RQ1.1., RQ1.2, and RQ1.3. The applications are categorized based on the sensors used, distinguishing between wearable systems and systems deployed in the environment, each further divided into subcategories. The applications in the “R1—Preventive” group act to keep the worker in the safe state (R1). The applications in the “R2—Reactive” group act to bring the worker back into the safe state as soon as possible. The applications in the “R3—Post-Incident” group act to minimize the damage after an incident occurred. The “R” stands for “risk state”. The references marked with an Asterix (*) are commercial applications. For further explanation, refer to the text.

Type of Sensor/Data Collected			
	R1—Preventive	R2—Reactive	R3—Post-Incident
Wearable Systems			
Physiological Monitoring	[14,15,16,17]	[15,17]	
Environmental Monitoring	[14,15,16,17,18,19]	[15]	
Movement and Posture Monitoring	[16,17,18,19,20,21] [22] * [23]	[17] [22] * [23]	[13,21,24] [22] *
Proximity and Location Tracking	[14,16,21,25] [22] *	[21,25] [22] *	[13,21] [22] *
Systems Deployed in Environment			
Image Sensors	[21,26,27,28,29,30,31] [32] * [22] * [33] * [34] * [35] * [36] * [37] * [38] * [39] * [40] * [23]	[32] * [22] * [33] * [34] * [35] * [37] * [38] * [39] * [40] * [23]	[32] * [22] * [33] * [34] * [35] * [37] * [38] * [39] * [40] *
Environmental Sensors	[16,17,19,41,42]	[17,41,42]	
Motion and Proximity Sensors	[19] [22] * [35] * [37] * [39] * [40] *	[22] * [35] * [37] * [39] *	[22] * [35] * [37] * [39] *

### 3.2. Findings for RQ1.1: Which AI-Based Wearable Systems and Technologies Are Implemented for Occupational Safety in Industrial Settings?

AI-based wearable systems designed to enhance occupational safety are often composed of devices that can be carried on the body and integrated into clothing or equipment, such as smart watches, smart PPE, etc. Their major advantages can be portability, user proximity, real-time applicability, energy autonomy, and discrete use. Enhanced with data storage, processing, and communication capabilities, these devices can be potential parts of integrated systems or IoT deployments. Primary applications of AI-based wearable systems for occupational safety include, but are not limited to, motion and activity detection, recognition of work-related musculoskeletal disorders, fall detection, evaluation of exposure to different physical agents, evaluation of exposure to chemical agents, the location of potential hazards, etc. [43]. We categorized the applications making use of wearable sensors according to the type of monitoring, i.e., (i) physiological monitoring, (ii) environmental monitoring, (iii) movement and posture monitoring, and (iv) proximity and location tracking (see Table 1). Almost none of the applications made use of only one sort of sensor unit, i.e., multiple sensors were integrated. Especially the wearable environmental monitoring category was a cross-cutting feature in all found systems, with environmental sensors being combined with physiological monitoring, movement and posture monitoring, as well as proximity and location tracking.

#### 3.2.1. Physiological Monitoring

Wearable systems that integrate components for physiological monitoring, measuring vital parameters, and biomedical signals, have significant potential for providing real-time health and safety insights not only in medical and sports domains but also in occupational settings. They enable early detection of health issues, stress, and fatigue, allowing for timely interventions and reducing the risk of accidents and long-term health problems. Heart rate measurement is a common feature across the majority of the studied cases, and often physiological monitoring goes in combination with other sensor units, such as environmental sensors. One example is given by Donati et al. [14], where they develop a comprehensive monitoring platform that consolidates several functionalities into one single tool. This platform simultaneously tracks workers’ vital parameters through a sensorised T-shirt (measuring e.g., ECG signals, respiracy parameters, blood pressure and movement of subject with inertial sensors), which can be worn under the work uniform without impacting the worker’s mobility and safety in real time, thus providing insights into their health status, stress, and fatigue levels. At the same time, data from production processes (e.g., process timelines), environmental conditions (e.g., temperature, humidity, brightness), and indoor localization are integrated into one cohesive system. Salahudeen et al. [15] develop a smart PPE for the mining industry, which tracks and detects potentially harmful circumstances in real time, exploiting the connectivity and longevity advantages of the Long-Range Wide-Area Network (LoRaWAN) technology. Besides environmental sensors (e.g., wearable gas sensor, temperature and humidity), the PPE integrates a pulse rate sensor, monitoring the person’s general health state while working. Similarly, Moe et al. [16] present an IoT-based safety system for the outdoor construction environment using a federated learning approach, collecting and analyzing worker data (heart rate) along with the weather data (e.g., wind, precipitation, temperature) and data of the building (i.e., the position of the operator relative to the building such as the slope). Another IoT-enhanced setup for workplace safety across various sectors is conceptualized by Raman et al. [17]. Their comprehensive system includes several environmental sensors and wearables, while the wearables monitor workers’ vital signs and mobility (heart rate sensors for health issues, stress, or overexertion detection). Additionally, wearable accelerometers assess ergonomic hazards by identifying dangerous movements. The system aims to prevent injuries by tracking repeated movements, vibrations, and sudden impacts. In general, the focus goes towards the simultaneous integration of several sensor units and a more holistic assessment of workers’ health and safety, in which real-time data processing and monitoring play an increasingly important role, with wearable systems enabling immediate feedback on, e.g., stress levels.

#### 3.2.2. Environmental Monitoring

As the operator is working under different environmental conditions, factors such as, e.g., the temperature, humidity, or brightness of the surroundings might have a high impact on the well-being of the person and thus, on occupational safety. The well-being of an operator is crucial for maintaining safety, as physical and mental discomfort can lead to reduced concentration, slower reaction times, and an increased risk of errors. Prolonged exposure to unfavorable conditions may cause fatigue, stress, or physical strain, all of which heighten the likelihood of workplace accidents. Sensor data become more valuable when integrated from multiple sensor units. Being a cross-cutting feature, environmental sensors will be addressed in each of the sections accordingly.

#### 3.2.3. Movement and Posture Monitoring

Wrong, non-ergonomic movements and postures can often be a reason for work-related musculoskeletal disorders. Wearable technologies, often leveraging inertial measurement units (IMUs), have the potential to track and analyze movement patterns, such as in the work by Astocondor et al. [18], who develop a wearable IoT-based ergonomical device for spinal posture rectification, operating with IMUs but also featuring integrated environmental sensors for temperature and humidity. The authors of Sangeethalakshmi et al. [19] examine unsafe ergonomic factors and develop a chair with integrated pressure mats to determine an ergonomic sitting position. Additionally, they use IMUs (accelerometers and gyroscopes) to collect movement data. The environmental parameters they use are temperature, humidity and luminance. Another example for movement and posture monitoring is presented by Rudberg et al. [21]. Here, they test different applications of AI in a real-world construction environment; among them, a smart helmet. The smart helmet contains a wearable IoT device that collects and reports movement and localization data to a digital twin. If a fall or hit is detected or if the person is in a danger zone, an internal alarm is triggered. Khan et al. [23] use an IoT-based approach with IMUs and altimeters to monitor workers climbing scaffolding at construction sites. Deploying sensor fusion, IMU and altimeter data are activated only when a worker is detected in a danger zone by visual recognition. Above a certain height, workers must attach their hook to the scaffolding. The status of this hook (attached/unattached) is monitored via IMUs (measuring acceleration, gyroscope, and magnetometer data) and pressure sensors, which assess acceleration, vertical velocity, angular velocity, tilt angle, direction, and pressure to identify unsafe behavior and hook status at risk heights. Bonifazi et al. [24] specialize in fall detection recognition. They design a personal device, called the sensor tile, carried close to the body on the waist, equipped with an IMU and a machine learning core component, making it possible to implement AI algorithms directly on the sensor without needing a processor and a Bluetooth module. This device is able to classify “not fall activities” (walking, running, sitting, stumbling without falling, etc.) and “fall activities” (e.g., tripping and falling, fainting). Connected to a so-called safety coordination platform that monitors the working environment, incidents can be communicated and managed. Narteni et al. [20] provide an example of using movement monitoring for fatigue detection. A deteriorated physical condition can increase the risk of casualties, injuries, slips, and falls during work. Their study addresses the problem of physical fatigue detection by comparing different models that monitor human movements using inertial measurement units (IMUs) placed at the ankle, hip, wrist, and chest of the participants. These non-invasive sensors measure acceleration, velocity, and orientation to assess fatigue levels. Regarding market applications, Trio Mobil [22] utilizes technologies with IMU sensor units specifically designed for lone worker safety, particularly for fall detection.

Overall, the findings in this section highlight the potential of wearable technologies, particularly those leveraging inertial measurement units (IMUs), in monitoring and improving workplace ergonomics, posture, and movement patterns. The studies demonstrate a wide range of applications, from posture correction and ergonomic assessment to fall detection and fatigue monitoring, often incorporating environmental data like temperature and humidity. These technologies might play an important role in preventing work-related musculoskeletal disorders and improving worker safety in high-risk environments such as construction sites.

#### 3.2.4. Proximity and Location Tracking

Proximity and location tracking, often implemented through GPS, RFID tags, or Bluetooth beacons, is commonly integrated with other sensor units in wearable devices. These systems are designed to enhance worker safety by providing real-time monitoring of personnel movement, ensuring compliance with safety protocols, and preventing accidents. They are crucial for determining whether a worker is in a danger zone or at risk of a collision, allowing for timely alerts and interventions. Here are some examples of implementations in the literature: Bong et al. [13] intend to design a system which relies mainly on activity tracking combined with indoor localization, while they realize activity tracking with IMUs and indoor localization with the development board switching connections to the closest access points (APs) nearby. Donati et al. [14], among other technologies in their comprehensive system, implement a subsystem of Real Time Location System (RTLS) tags in order to provide real-time spatial positions of tagged objects such as PPE or assets. They do not explicitly specify the underlying technology but indicate that they use middleware for data exchange, using the JSON format for it. Moe et al. [16] use a handheld GPS in their worker safety prediction mechanism to localize workers on construction sites, enabling quick location in incidents and prevention of entry into danger zones. Rudberg et al. [21] deploy Bluetooth beacons with the IoT device in their smart helmet for localization. Kim et al. [25] implemented a smart helmet-based proximity warning system to enhance occupational safety on the road. This system utilizes image sensors with a camera module for proximity and location tracking. An advantage of this approach is that it can recognize all objects defined as hazardous and not only objects that are tagged prior to this. This is the only application using an image sensor but we categorized it into the wearables category because the camera is integrated into a smart helmet and not mounted in the environment. Regarding market applications, Trio Mobil [22] deploy RTLS (using several technologies such as ultra-wideband (UWB), Bluetooth Low Energy (BLE), or GPS) for industrial IoT, the tracking of operators, and vehicles.

### 3.3. Findings for RQ1.2: Which AI-Based Systems and Technologies Deployed in the Environment Are Implemented for Occupational Safety in Industrial Settings?

AI-based systems with environmental sensor components are commonly used to oversee larger work areas, such as shop floors, and monitor the safety compliance of multiple individuals simultaneously. In these use cases, we identified three recurring categories of sensors: (i) image sensors (e.g., computer vision), (ii) environmental sensors (e.g., continuous air quality monitoring), and (iii) motion and proximity sensors, which are often integrated with image monitoring systems.

#### 3.3.1. Image Sensors

Image sensors, which detect and extract information like light or infrared radiation to form images, make up the largest group of research papers in this study, if we look across research and commercial applications. All the commercial applications examined in this research fall into this category. Here are some examples of how they are used: Rudberg et al. [21] test three AI image sensor implementations in a real-world construction scenario. They implement IP RGB cameras (Internet protocol Red-Green-Blue cameras: a network-connected camera that captures and transmits full-color video or images over the internet or a local network) at the gate to check whether operators wear proper safety equipment (HVC, helmet, helmet chinstrap, goggles) using computer vision and object detection. These are also applied with IP cameras on a tower crane jib to observe, detect, and track objects on the ground in real time, providing alerts to the crane operator if workers enter a defined safety zone. The third application involves an IR camera unit on a heavy machine, detecting human workers nearby by recognizing the reflective properties of warning clothes, and alerting both the driver and the worker to the risk.

Since helmets are mandatory widely adopted safety equipment, helmet recognition is a frequently researched application. Anjum et al. [28] develop a computer vision-based mobile solution for recognizing safety helmets at construction sites. This system uses object detection to determine whether a worker is wearing a helmet and sends an alarm message to the safety supervisor if they are not. It can be deployed on edge devices, such as smartphones, and includes a real-time database that records workers’ safe and unsafe activities for future analysis. Similarly, for power line safety inspection, Bian et al. [29] develop a safety helmet detection mechanism that relies on drone images to detect whether workers are wearing helmets. Ji et al. [31] also investigate safety helmet recognition, introducing a hybrid attention module.

Khan et al. [23] enhance the safety of construction workers climbing scaffolding by deploying computer vision (CCTV) and IMU sensor fusion, where one sensor unit only starts working after the other sensor unit recognizes it is necessary. Ran et al. [26] develop a device that automatically detects safety signs near individuals in a chemical manufacturing plant. They test various detection models and establish a benchmark dataset for chemical safety signs, focusing extensively on the dataset rather than the application. The system assists operators in “seeing” safety signs and proactively draws their attention towards them. Yuan et al. [27] create a monitoring system for an electric power construction site aimed at preventing electric shock and falls. Their system can detect a variety of situations, including personnel identification, violations (e.g., improper PPE usage, smoking), environmental abnormalities (e.g., fireworks), equipment status, and safety tool identification. The videos are processed in real time, and early warning information is broadcast to the responsible individuals or workers. Fang et al. [30] investigate a system for recognizing protective masks in industrial scenarios, utilizing deep learning and AI edge computing.

As to the commercial applications investigated for this research, all 10 companies and applications we looked at make use of image sensors and computer vision, utilizing these advanced AI-based technologies aimed at improving safety in industrial settings. Eight of the commercial applications specialize on safety surveillance ([22,32,33,35,37,38,39,40]), two specialize on ergonomics ([34,36]). Trio Mobil [22] has created a real-time computer vision system that uses image processing, deep learning, and edge processing to monitor and control pedestrian and vehicle interactions, with applications such as dock, crane, and machine zone safety. Intenseye [32] focuses on real-time monitoring of industrial facilities, while Chooch [33] specializes in workplace safety through vehicle control, access control, PPE compliance, and behavioral safety monitoring. Eyyes [35] develops software and hardware for safety-critical AI applications in object classification and detection across industries like automotive, railway, transport, and manufacturing. Nirovision [37] offers monitoring and object detection solutions, and Geutebrück [38] provides intelligent video surveillance with object identification, anomaly detection, and access control features. ProtexAI [39] uses computer vision to help companies monitor unsafe behaviors, and AssertAI [40] focuses on AI-based surveillance for manufacturing and other industries. 3motionAI [34] specializes in motion analysis and ergonomics, while EWIworks [36] offers solutions for workplace ergonomics, including its PoseChecker software (v.1), which aims to prevent musculoskeletal disorders through video-based motion capture.

#### 3.3.2. Environmental Sensors

This group includes applications that use environmental sensors which measure, e.g., temperature, humidity, lighting conditions, and air quality. Unlike wearable sensors, they are mounted in the environment, although the placement of some sensors is not always clearly specified in the papers, especially in contexts of IoT-setups, as they combine several sensor units.

Arun et al. [41] demonstrate the potential of AI in enhancing hazard assessment and safety measures in petrochemical industries by using sensors for temperature, pressure, pH, and vibration connected to an Arduino board for real-time hazard detection and alarm features. Moe et al. [16] present an IoT-based safety system for outdoor construction that collects and analyzes weather data such as wind, precipitation, and temperature. Raman et al. [17] employ sensors for temperature, humidity, air quality (CO2, VOC), and noise levels in their IoT-enhanced workplace safety system, using a rule-based approach with alarms triggered by threshold breaches. Sangeethalakshmi et al. [19] incorporate IMU sensors for ergonomic analysis along with sensors for temperature, humidity, and light, some of which are mounted in the environment. Chang et al. [42] develop a predictive model to measure respirable dust (tiny airborne particles, typically less than 10 microns in diameter, that are small enough to be inhaled into the lungs, posing health risks such as respiratory diseases when exposed over time) in the workplace, combining low-cost sensors (CO_2_, PM_2.5_, PM_10_, temperature, and relative humidity) into a respirable dust sensor module using various AI algorithms.

#### 3.3.3. Motion and Proximity Sensors

While this category can be considered as a subset of image sensors, it focuses specifically on applications that measure motion and proximity. It includes sensors like LIDAR (Light Detection and Ranging), radar, and infrared sensors, which are designed to detect and analyze movement and distance with high precision. Sangeethalakshmi et al. [19] use an infrared sensor for detecting motion for their use case where they analyse ergonomic factors and design a chair that can analyse ergonomic sitting positions. Commercial applications such as Trio Mobil [22], Eyyes [35], Nirovision [37], ProtexAI [39] and AssertAI [40] all deal with use cases which imply motion and proximity detection with sensors deployed in the environment or attached to machinery. For example, Eyyes implements safety-critical applications for object classification and detection for automotive, railway, and transport, inferring the recognition of humans and vehicles for motion detection and collision avoidance. In general, the findings indicate that motion and proximity detection technologies, including LIDAR, radar, and infrared sensors, play a crucial role in both research and commercial applications, helping to avoid collisions by accurately detecting and responding to human and object movements in various industrial contexts.

### 3.4. Findings for RQ1.3: Which Preventive, Reactive, and Post-Incident Features of These AI-Based Safety Applications Can Be Identified?

The design of a holistic workplace safety system should integrate three key parts—injury prevention, the detection of a hazard, and injury response [13]. In this part of our paper, we adopt a simplified version of the framework by Pishgar et al. [3], where they investigate how AI systems can be used in detecting, preventing, and controlling the evolution of safety accidents. For the classification of papers, see Table 1. A worker can be in different levels of safety risk at any given time—we define these states as R1, R2, and R3 (where the R stands for risk state). R1 is the ideal state where a worker has minimal risk of exposure to the hazard. The goal is to keep the worker in this state. In R2, the worker is at an increased risk of a harmful work-related exposure event but has not experienced such an event (yet). Here, the goal is to bring the worker back into R1 as soon as possible. R3 is the state when a harmful work-related event has already occurred, impacting the health and safety of the worker. Here, the goal is to minimize the damage to the worker.

We have analyzed the systems described in the papers according to their stage of intervention–if they operate to keep the worker in the safe state, we categorized them as “preventive”. If they operate in such a way that they detect whether the worker is in R2, that is, the hazardous state, and work to bring him back into the safe state, we categorized them as “reactive”. If the systems have the potential to minimize damage after an incident has occurred, they go into the “post-incident" category.

#### 3.4.1. Preventive Features

Most applications fall into this category since their primary implementation is to keep the worker in the safe state. Some of the systems feature elements that qualify them for the other two categories, in which case papers appear several times in Table 1. In our framework, the difference between preventive and reactive features lies largely in the ability of reactive features to react and give off an alarm if the operator is not in the safe state anymore. Preventive systems do not necessarily do this, e.g., a preventive system can be simply an assistance system which helps the operator following a protocol of worksteps. In this case, it might be said that the assistance system indirectly supports safety as, e.g., the operator’s stress levels are lower due to the assistance and thus he is less likely to be inflicted harm by work overload. In this case, we could say, this is only a preventive feature.

In the literature, keeping the worker in the safe state is often realized by creating comprehensive real-time monitoring IoT-based infrastructures with multiple sensor units, partly worn on the body, partly deployed in the environment, monitoring health or environmental conditions and providing insight for the operator via user interfaces ([14,15,16,17]). Another preventive application with IoT-based mechanisms is the deployment of IMUs for posture and movement monitoring, where the system draws the operators’ attention to the fact that a person does not move in an ergonomically sustainable way, such as in [18], where they design a posture-correction wearable device attached to the back. Similarly, a smart posture-correction chair with integrated pressure sensors was implemented in [19]. Narteni et al. [20] used IMU data for the analysis of physical fatigue, demonstrating that timely fatigue detection is a preventive measure. In the context of systems deployed in the environment, image sensors, particularly those performing preventive functions, represent the largest category. A common preventive use of these sensors is the detection of PPE and helmet compliance [21,26,27,28,29,30,31], as they monitor whether operators are wearing the required protective gear. In the work of Khan et al. [23], computer vision is utilized to detect whether workers are entering a danger zone. In the work of Yuan et al. [27], preventive measures extend to monitoring all worker behaviors to enable real-time intervention if they engage in dangerous activities, such as smoking. All ten commercial systems examined in this paper employ computer vision for preventive purpose for PPE compliance, ensuring adherence to safety protocols and keeping workers away from danger zones.

#### 3.4.2. Reactive Features

Preventive and reactive features in safety systems often complement each other. A system designed to maintain a worker’s safety frequently includes mechanisms to detect when the worker has exited a safe state, triggering an alarm to alert the worker or safety personnel of the hazard. For this category, we have included systems that notify the operator or relevant personnel in any way, enabling them to intervene and reduce the risk to the worker. Real-time notifications via a smartphone app or UI are used in [15,17], where hazardous circumstances trigger quick alerts automatically. In the work of Arun et al. [41], an IoT system is equipped with a buzzer that sounds an alarm when any of the sensor readings exceed the maximum limit, providing a visual and auditory alert. In the work of Rudberg et al., [21] an alarm goes off when a worker enters a restricted zone wearing the IoT smart helmet. Similarly, unsafe behavior (not attaching a hook while climbing on scaffolding) induces an alarm in Khan et al.’s work [23], in order to make the worker change his behavior and thus return to the safe state. The authors of Khan et al. use multisensor fusion, combining computer vision and IMUs, achieving better detection of unsafe behavior. Other reactive features in the image sensor category are found in commercial applications, which trigger alarms when hazardous events occur. For instance, Intenseye [32] uses an AI-powered computer vision technology to connect with smart devices and IoT equipment, providing immediate alerts for unsafe behaviors or body mechanics. Similarly, other commercial applications offer software that allows for user interaction, providing comprehensive packages with insights and analysis tools for safety administrators.

#### 3.4.3. Post-Incident Features

Post-incident features are tools and mechanisms designed to limit and control the damage after an incident occurred. Automated injury detection using AI can help by quickly identifying injuries, sending crucial information to first responders, and alerting nearby staff, which reduces delays and improves emergency response. Relying on a third person to report an injury can cause delays, especially in the case of isolation, making automated systems a safer option for quick injury detection and response. A frequent AI-based application of a post-incident feature in industrial domains is fall detection, since a fall is an unwanted, hazardous incident after which the person must be found and provided with help immediately.

In the wearable system category, post-incident features have been identified only in the “Movement and Posture Monitoring” and “Proximity and Location Tracking” groups, with all examples focused on fall detection. Bong et al. [13], using an IoT IMU node attached to a worker’s limb, propose an injury response system that generates a notification after a fall. This notification includes the worker’s identity, the type of detected injury pattern, the location, and the expected nature of the injury based on that location. In the work of Rudberg et al. [21], a smart helmet equipped with an IoT-IMU device is implemented for fall detection. However, during the study, which also examined the usability of the applications, it was noted that, due to the device’s heavy weight, workers attached it to other parts of their bodies, resulting in false fall detections. Bonifazi et al. [24] implemented and tested wearable and area devices for fall detection. They established a safety coordination platform for fall detection, where they suggested to combine wearable devices with IMUs and video analysis, suggesting that wearable devices and devices mounted in the area can be leveraged as mutual validators. All commercial applications analyzed in this paper incorporate computer vision and image sensors to provide post-incident features. The advantages might be, e.g., that these systems offer immediate assistance and mitigation, collect data for safety improvements to prevent future accidents, and can support legal proceedings.

### 3.5. Findings for RQ2: Given That Some AI-Based Applications for Occupational Safety Inherently Introduce New Risks While Others Do Not, Does the Respective Scientific Literature Address These Risks? Additionally, When Considering Both Explicitly Addressing Risks and Those Not Posed by the Design of the Respective Systems, Which Applications Can Be Considered Safer or Less Safe Overall?

One of the key contributions of this paper is the anticipation of challenges associated with the deployment of AI-based systems for occupational safety, which may pose risks to users. To address this, we examined potential harms these systems could present to operators and explored approaches for conducting effective risk evaluations. After reviewing several regulatory guidelines for AI worldwide, at this point we would like to mention the Ethics Guidelines for Trustworthy AI [44] published in 2019. These guidelines provide a framework for Trustworthy AI, focusing on two of its three components: ethical AI, ensuring adherence to ethical principles and values, and robust AI, both technically and socially, since even well-intentioned AI systems can cause unintentional harm [44]. The third component, lawful AI, is addressed by the European Union’s AI Act, the first legal regulation for AI globally which was adopted in April 2024. Furthermore, the European Commission has called for the development of “harmonized standards”, to support this legal framework. The ISO/IEC 23894:2023 standard (Information technology—Artificial intelligence—Guidance on risk management) [45] currently serves as one of the main references for risk evaluations in AI systems, providing a key framework while the broader regulatory landscape is still under development. When identifying risks associated with AI systems, various AI-related objectives must be considered, depending on the system’s nature and application context. These objectives, outlined in Annex A of the ISO/IEC 23894:2023, include but are not limited to the eleven listed below. While initially designed for organizational use, they are highly relevant for our scope as they summarize key factors that need to be addressed when undergoing an AI risk analysis:Fairness: AI systems for automated decision-making can produce unfair outcomes due to biases in data, objective functions, and human input. Additionally, unfairness can arise from biases in the design, problem formulation, and deployment decisions of AI systems.Security: In AI, particularly in machine learning systems, new security issues such as data poisoning, adversarial attacks, and model stealing must be addressed in addition to traditional information and system security concerns. These emerging threats pose unique challenges that go beyond classical security measures.Safety: Safety relates to the expectation that a system does not, under defined conditions, lead to a state in which human life, health, property, or the environment is endangered. Use of AI systems in automated vehicles, manufacturing devices, and robots can introduce risks related to safety.Privacy: Privacy involves individuals controlling how their information is collected, stored, processed, and disclosed. Given that AI systems often rely on big data, particularly sensitive data like health records, there are significant concerns about privacy protection, potential misuse, and ethical impacts, including discrimination and freedom of expression.Robustness: Robustness is related to the ability of a system to maintain its level of performance under the various circumstances of its usage. In the context of AI systems, there are additional challenges due to its complex, nonlinear characteristics.Transparency and explainability: Transparency involves both the characteristics of organizations operating AI systems and the systems themselves. It requires organizations to disclose their use of AI, data handling practices, and risk management measures, while AI systems should provide stakeholders with information on their capabilities, limitations, and explainability to understand and assess their outcomes.Environmental impact: AI can impact the environment both positively, such as reducing emissions in gas turbines, and negatively, due to the high resource consumption during training phases. These environmental risks and their impacts must be considered when developing and deploying AI systems.Accountability: Accountability in AI involves both organizational responsibility for decisions and actions, and the ability to trace system actions to their source. The use of AI can change existing accountability frameworks, raising questions about who is responsible when AI systems perform actions.Maintainability: Maintainability is related to the ability of the organization to handle modifications of the AI system in order to correct defects or adjust to new requirements. Because AI systems based on machine learning are trained and do not follow a rule-based approach, the maintainability of an AI system and its implications need to be investigated.Availability and quality of training and test data: AI systems based on machine learning require high-quality, sufficiently diverse training and test data to ensure intended behavior and strong predictive power. The data must be validated for currency and relevance. The amount will vary based on intended functionality and environment complexity.AI expertise: AI systems require inter-disciplinary specialists (difference to traditional software solutions) for their development, deployment, and assessment. End users are strongly encouraged to familiarize themselves with the functionality of the system.

To explore RQ2, we revisited the 20 selected research publications to evaluate whether the associated risks had been adequately addressed in each paper. Due to some gaps in provided information, we conducted a best practise analysis. The findings of this analysis are shown in Table 2. It is important to note that the degree to which the identified objectives were addressed does not reflect any judgement of the overall quality of the papers. Instead, our focus was on understanding two key aspects: (i) whether the papers addressed the associated risks, and (ii) in cases where the risks were not addressed, whether we, in our assessment, considered those risks significant enough to be addressed.

Fairness and environmental impact appear not to have been addressed in any of the papers, followed closely by frequent omission of addressing safety, maintainability, AI expertise, security, and privacy.

Fairness: Systems can become unfair due to biases in data, objective functions, human input, or even in the design, problem formulation, and deployment decisions. Notably, we noted that the availability and quality of training and test data is addressed in nearly every paper. This suggests that the authors highlighted the importance of high-quality and sufficient data in their research. We might make the assumption that, by doing so, authors implied that this includes fairness. However, fairness is a complex issue that extends beyond ensuring diverse and representative data. Mitigating bias can also be achieved through, e.g., customizable features that allow users to tailor the system to their needs, inclusive design that considers different user abilities and backgrounds, accessibility measures such as alternative input methods or screen reader compatibility, active user feedback integration to identify and address biases, and diverse development teams that bring multiple perspectives to the design and evaluation process.

Environmental impact: The environmental impact of AI-based systems, such as the high resource consumption during training phases, was not addressed. While it may be debatable whether this objective falls directly within the scope of each study, it is important to note that this issue is highlighted in guidelines for safe and trustworthy AI. Despite its inclusion in these frameworks, our research shows that it does not yet receive attention explicitly.

Safety: In the context of our study, where we investigate whether AI-based systems which enhance occupational safety might pose an additional risk to the (in this case, physical) safety of the users themselves—e.g., collisions—safety is a challenging issue. According to the norm we apply in this context, systems which can be dangerous for the safety of a user typically include self-driving vehicles, applications of robotics, or manufacturing devices. By this definition, most of the applications in the research we are conducting do not pose a risk to the user, as mainly we are looking at systems that are IoT setups or computer vision systems. However, there are exceptions, such as in the work of Bian et al. [29], where they use camera-equipped drones for monitoring safety. Additionally, in the papers, the words safety and security were often used by the authors as synonyms. This is misleading for the analysis, since safety refers to ensuring that AI systems operate without causing unintended harm, while security focuses on protecting the systems from malicious attacks or unauthorized access.

Maintainability and AI expertise: Both objectives, maintainability and AI expertise, received identical checkmarks and crosses across all the papers reviewed. This may be because, in our analysis, they often appeared closely related or even synonymous in certain situations. For instance, AI expertise is essential in designing modular, scalable, and adaptable systems, ensuring the architecture is maintainable and can be easily updated, modified, or expanded. Ensuring system maintainability requires strategies like regular performance monitoring, which must be carried out by personnel with specialized AI expertise. The AI expertise objective emphasizes the need for interdisciplinary specialists involved in the development, deployment, and assessment of AI systems. Additionally, it encourages users to gain a deeper understanding of how the system functions, making collaboration and knowledge transfer between experts and users essential. In fact, whenever maintainability was addressed, AI expertise was similarly discussed. For example, Kim et al. [25] propose vision-enhanced smart helmets for object recognition, highlighting that AI systems need regular monitoring to maintain performance over time. While the authors offer valuable insights into the potential of their system and transparently acknowledge certain limitations, including maintainability, they do not propose specific solutions for overcoming these challenges. This is why we marked both maintainability and AI expertise with a cross for this paper and others with similar gaps. In contrast, Rudberg et al. [21] demonstrate how an iterative, agile approach proved effective during the pilot testing of AI systems. Their methodology not only improved maintainability but also addressed the need for AI expertise, showcasing how an agile process can actively integrate both aspects to ensure long-term system success.

Security: Security is fundamentally about preventing malicious actors from gaining access to a system or its components. Manipulations of models or data inputs can undermine the system’s reliability and pose significant risks. For instance, unauthorized access can lead to privacy violations, allowing for sensitive information to fall into the hands of unauthorized individuals. It is important to mention the difference between security and privacy: while security focuses on protecting systems from unauthorized access or attacks, privacy is concerned with controlling who has access to sensitive information and how it is used. Moe et al. [16] propose a federated learning (FL) approach in outdoor construction environments, highlighting its robust security features that enhance data privacy by eliminating the need to transmit local data to a central server during the training process. They emphasize that preserving the privacy and security of sensitive data is a primary reason for their choice. In contrast, Donati et al. [14], while designing and testing a comprehensive safety monitoring system, analyzing workers’ stress and fatigue data from the environment and production processes, do not explicitly address security measures in their paper.

Privacy: The applications can be broadly classified into either a form of IoT setup or a form of computer vision setup (some papers like that of Rudberg et al. [21] deploy both). For an application to respect users’ privacy, it must ensure that individuals can control how their data are collected, stored, processed, and disclosed. Notably, there is a distinct divide between IoT-based and computer vision-based applications in this regard, shown by mostly checkmarks in IoT applications and mostly crosses in vision-based systems. In several IoT setups, the choice of technology, such as using IMUs for activity recognition, was supported by the argument that it better respects user privacy, since, e.g., vision-based motion capture systems have issues with privacy and occlusion [19]. At the same time, in most of the applications utilizing computer vision monitoring, user privacy was either not addressed or mentioned as a limitation (e.g., Kim et al. [25]). One exception is the work of Khan et al. [23], where the system uses computer vision to detect when workers exceed a specific height and enter a danger zone. In this case, privacy is addressed by avoiding identification and focusing solely on monitoring the workers’ status.

## 4. Discussion, Limitations, and Future Work

### 4.1. Summarizing Findings

#### 4.1.1. RQ1.1

In summary, AI-based wearable systems for occupational safety are composed of devices integrated into clothing or equipment or worn on the body. We categorized these systems based on the type of monitoring and data collection: (i) physiological monitoring, (ii) environmental monitoring, (iii) movement and posture monitoring, and (iv) proximity and location tracking. Most applications utilize a multisensory approach, frequently within IoT setups, with a focus towards developing comprehensive safety monitoring tools. For instance, environmental monitoring—typically tracking parameters such as temperature, humidity, brightness, etc.—is never standalone but, in the papers we analyzed, always combined with other sensor units to supplement other data. Physiological monitoring focuses on detecting health issues, stress, or fatigue, with heart rate being the most commonly measured parameter, alongside others such as respiratory parameters and blood pressure. Movement and posture monitoring, often using inertial measurement units (IMUs), can assess ergonomic and non-ergonomic movement patterns and postures, detect fatigue and falls. Proximity and location tracking is frequently implemented via GPS, RFID, or Bluetooth beacons, to monitor workers in danger zones, localize assets, or to prevent collisions.

#### 4.1.2. RQ1.2

In contrast to wearable systems, systems deployed in the environment are used to monitor larger work areas for safety compliance, such as in construction or large manufacturing halls, and more individuals at once. We identified three categories: (i) image sensors (e.g., computer vision), (ii) environmental sensors (e.g., continuous air quality monitoring), and (iii) motion and proximity sensors, which are often integrated with image monitoring systems. Common applications for image sensors monitoring are safety equipment detection (especially helmets), access control, identification of personnel in danger zones, and collision avoidance. Some systems are trained to observe and detect behaviors that conflict safety protocols (e.g., smoking at an electric power construction site) or environmental abnormalities. Safety surveillance and ergonomics are the most common computer vision applications on the market. Environmental sensors deployed in the environment typically measure parameters that can affect the health and well-being of operators such as temperature, humidity, pressure, lighting conditions, air quality, etc. Motion and proximity sensors such as LIDAR, radar or infrared, designed to accurately detect movement and distance and often integrated with image sensors, are commonly used in commercial applications for collision avoidance.

#### 4.1.3. RQ1.3

A holistic AI-based safety system can be designed to address all three aspects of occupational safety: injury prevention, detection of hazards, and injury response after an incident. In most cases, systems have features that belong to the preventive and reactive categories, and in some cases, the post-incident category. It was challenging for us to make a clear distinction between the preventive and reactive categories in systems. Preventive features include, for example, when a system can monitor factors that enhance safety—such as an assistance system that helps with workflow and thus influences safety indirectly, a system that can monitor PPE compliance and detect if an operator is not wearing a helmet, or an assistance system that helps recognize chemical safety signs. According to our definition, these features work to keep the operator in a safe state. Reactive features, however, aim to bring the operator back to a safe state whenever the system detects they are no longer in one. This could include sounding an alarm and requiring the operator to change their behavior to avoid a collision. Post-incident features involve anything that minimizes harm to the operator, such as fall detection, localization of the injured person, and immediate alerting of helpers. Such applications are still very rare in comparison to preventive and reactive applications, which might highlight an important research gap.

#### 4.1.4. RQ2

Since the first step in designing safer and more trustworthy AI systems or evaluating their risks is to understand which objectives must be met, we analyzed whether the scientific literature addresses these risks. After selecting the ISO/IEC 23894:2023 standard (Information technology—Artificial intelligence—Guidance on risk management) [45] as our framework for necessary objectives, we revisited the articles and identified notable differences across the objectives. Of the 11 objectives outlined in the framework, fairness and environmental impact were not explicitly addressed, while safety, maintainability, AI expertise, security, and privacy were frequently overlooked. For activity recognition, IMU-based technologies are potentially more privacy-preserving than computer-vision technologies. Although the papers varied in scope and research focus, and it can be debated whether these objectives could have been adequately addressed at the specific stage of each study, it is crucial to recognize these gaps. This highlights an important contribution to the ongoing conversation about the design of AI safety systems and the need for more comprehensive risk management.

### 4.2. Limitations and Future Work

This research stretches along a wide range of topics: (i) the categorization of AI-based technologies for occupational safety in industrial settings, with a specific emphasis on human-centered applications and sensor technologies; (ii) the exploration of the operational dimensions of these applications, classifying them into preventive, reactive, and post-incident categories; and (iii) an analysis of whether the objectives for trustworthy and safe AI-based systems are adequately represented and addressed in the respective literature. Given the inherent complexity of these subjects, it is challenging to address each topic in small detail. Consequently, revisiting the literature to conduct more detailed assessments, e.g., regarding security considerations, is desirable for future research. Additionally, investigating technical components beyond sensors from a more analytical perspective could lead to valuable insights.

Moreover, the research raises significant questions, particularly in contexts where strict monitoring of individuals is implemented alongside a zero-accident policy, as exemplified by Yuan et al. [27]. In such scenarios, concerns regarding privacy, explainability, and transparency emerge. While enhanced monitoring may substantially improve safety and reduce the likelihood of incidents, it is crucial to consider whether the resulting benefits to safety justify potential adverse effects on workers’ well-being, or whether the balance between these factors depends on the specific nature of the occupation (e.g., high-risk versus low-risk environments). Addressing these questions is essential for the development of ethically sound and effective AI-based safety solutions.

## 5. Conclusions

In this paper, we provided a comprehensive survey of AI-based safety systems in industrial environments, focusing on applications that enhance occupational safety by assisting or monitoring workers. We categorized these systems based on their preventative, reactive, and post-incident functionalities, and critically evaluated their underlying technologies. Additionally, we highlighted potential new risks these systems might inflict on the user.

By analyzing the current research landscape and examining both technological advancements and ethical considerations, we provided insights into the challenges and opportunities for designing AI systems that prioritize safety of operators in industrial settings. Future research should aim to address these challenges, ensuring that AI not only prevents accidents but also operates within a robust ethical framework.

## Figures and Tables

**Figure 1 ijerph-22-00705-f001:**
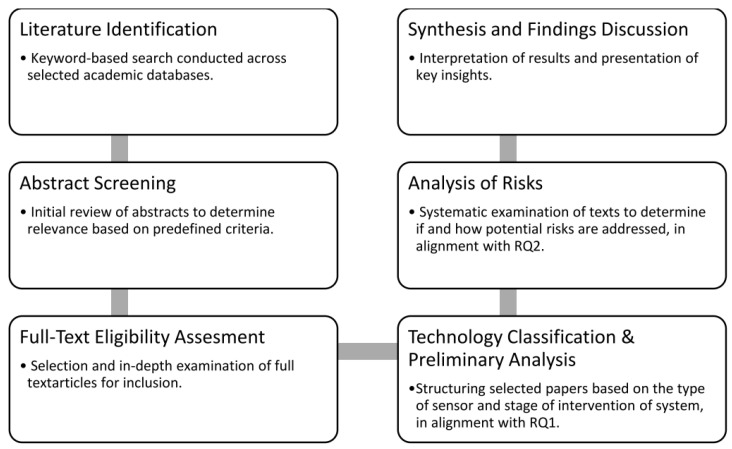
Research logic of this review.

**Figure 2 ijerph-22-00705-f002:**
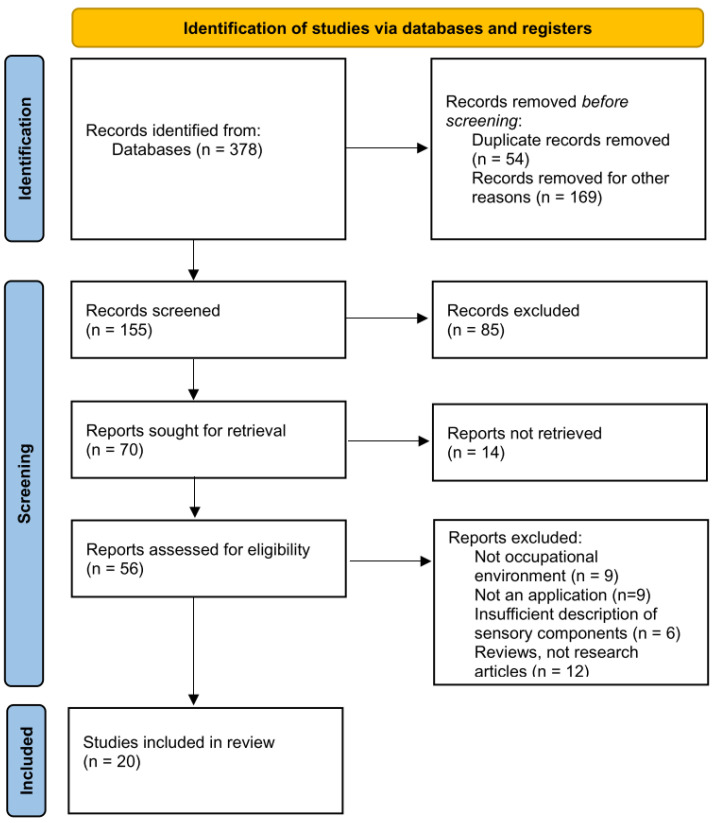
Process flow that demonstrates the research analysis in this study.

**Table 2 ijerph-22-00705-t002:** Objectives for AI risk management: If a topic was addressed in the respective paper, the corresponding box was marked with a checkmark. If it was not addressed in the paper, the corresponding box was marked with a cross. If the topic was not addressed in the paper but we, in our assessment, found it to be relevant to address, we marked it with a large cross. Additional remarks can be found in the text.

Paper	Fairness	Security	Safety	Privacy	Robustness	Transp./Explainab.	Environmental Imp.	Accountability	Maintainability	Data	AI Expertise	Key Goal
Bong et al. [13]				✓						✓		IoT-based injury detection via IMU activity tracking and localization
Donati et al. [14]			✓	✓	✓	✓		✓	✓	✓	✓	integrating activity and location data into one monitoring system
Salahudeen et al. [15]				✓	✓	✓		✓	✓	✓	✓	coal mine monitoring IoT
Moe et al. [16]		✓		✓	✓	✓		✓		✓		IoT at construction site using federated learning (FL)
Raman et al. [17]		✓		✓	✓	✓		✓		✓		IoT sensor network for workplace safety
Astocondor et al. [18]				✓	✓	✓		✓	✓	✓	✓	IoT biomechanical system for spinal rectification
Sangeethalakshmi et al. [19]		✓		✓	✓	✓		✓	✓	✓	✓	IoT human motion analysis, ergonomic smart chair
Narteni et al. [20]		✓		✓		✓		✓	✓	✓	✓	IMUs for fatigue prediction
Bonifazi et al. [24]			✓	✓	✓	✓		✓	✓	✓	✓	IMU wearable device for fall detection
Arun et al. [41]				✓		✓				✓		IoT system for safety in petrochemical industry
Chang et al. [42]				✓		✓		✓		✓		IOT low-cost sensor technology for estimation of respirable dust
Rudberg et al. [21]		✓		✓	✓	✓		✓	✓	✓	✓	five case studies in real world scenario
Khan et al. [23]				✓	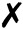	✓		✓		✓		fall prevention from scaffolding using IoT and vision sensor fusion
Kim et al. [25]		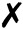	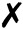	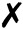	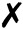	✓		✓	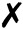	✓		smart helmet-vision-based proximity warning system
Ran et al. [26]				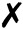	✓	✓		✓		✓		dataset creation for chemical safety signs detection mechanism
Yuan et al. [27]			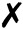	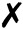		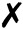		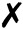		✓		rigidly monitoring system for electric power construction
Anjum et al. [28]				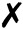	✓	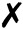				✓		safety helmet identification on edge device
Bian et al. [29]			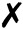	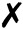	✓	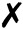				✓		safety helmet detection based on drone images
Fang et al. [30]				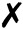	✓			✓		✓		safety mask monitoring based on edge computing
Ji et al. [31]				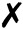	✓							safety helmet detection

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
