# Peer review of "Evaluating User Safety Aspects of AI-Based Systems in Industrial Occupational Safety: A Critical Review of Research Literature"

_ijerph, 2025, doi:10.3390/ijerph22050705_

Round 1

Reviewer 1 Report

Comments and Suggestions for Authors

This was a great add on to the work completed by Pishgar. I only have a few small comments related to some typos in the paper.

pg. 3 line 87 your sentence is incomplete.

pg. 4 lines139-145. This paragraph was confusing. I'm not sure why you did this search and I didn't see any further discussion about this throughout the rest of the paper.  Please either elaborate on the purpose of this web search or remove.

pg. 8 lines 255-260. This sentence is very long and confusing. It begins with "while wrong", but I was not sure what was wrong by the end of the sentence.

pg. 8 line 262 - IMUs has not yet been defined (it is spelled out on page 9 line 287 which is after this first use of the acronym. Please spell out here.

pg. 9 line 311 - the sentence "They do not explicitly specify the underlying technology in their but indicate that they use middleware..." the word their doesn't belong and should be substituted to make the sentence make sense.

pg. 10 line 338 - I do not know what an IP RPG camera is - can you spell out this acroynym.

pg. 13 lines 485 and 488 you are missing the name of the authors before the references (references 21 and 23). In line 489 you say the authors of "this" paper and it would be more clear if you just named the authors because I couldn't tell which authors you were referring to.

Author Response

Thank you very much for taking the time for making this review! I implemented all of the points:

Comment 1: “pg. 3 line 87 your sentence is incomplete.”

Response 1: Thank you for pointing this out – I fixed it in the text.

Comment 2: “pg. 4 lines139-145. This paragraph was confusing. I'm not sure why you did this search and I didn't see any further discussion about this throughout the rest of the paper.  Please either elaborate on the purpose of this web search or remove.”

Response 2: I have accordingly revised and modified on this comment. Particularly, this section refers to the commercial applications we examined in our paper to get a comparison between the non-commercial and the commercial applications. We return to these commercial applications a couple of times in the text, e.g., Table 1 (p. 7), these applications are marked with an asterix. Also we refer to these applications on p. 10, 369-387; p. 11 415-424.p. 13 520-521, and further occasions.

Comment 3: “pg. 8 lines 255-260. This sentence is very long and confusing. It begins with "while wrong", but I was not sure what was wrong by the end of the sentence.”

Response 3: Thank you for this comment, the sentence has been fixed in the text.

Comment 4: “pg. 8 line 262 - IMUs has not yet been defined (it is spelled out on page 9 line 287 which is after this first use of the acronym. Please spell out here.”

Response 4: IMUs are defined shortly before that, on page 8 lines 256-257.

Comment 5: “pg. 9 line 311 - the sentence "They do not explicitly specify the underlying technology in their but indicate that they use middleware..." the word their doesn't belong and should be substituted to make the sentence make sense.”

Response 5: Thank you, it is fixed.

Comment 6: “pg. 10 line 338 - I do not know what an IP RPG camera is - can you spell out this acroynym.”

Response 6: Thank you, I spelled out and explained the acronym now in the text.

Comment 7: “pg. 13 lines 485 and 488 you are missing the name of the authors before the references (references 21 and 23). In line 489 you say the authors of "this" paper and it would be more clear if you just named the authors because I couldn't tell which authors you were referring to.”

Response 7: Thank you, I spelled out the names of the authors in the text.

Reviewer 2 Report

Comments and Suggestions for Authors

This review article deals with AI-based systems in industrial settings aiming to enhance occupational safety. The authors have three objectives. Firstly, they categorize these systems regarding the type of sensor (wearable vs. environmental) and the type of data collected (e.g., images, biosignals, environmental parameters). Secondly, they distinguish between preventive, reactive, and post-incident applications. Thirdly, they analyse if the objectives for trustworthy AI as stated in the ISO/IEC 23894:2023 are represented in the reviewed papers. Finally, the authors conclude that the majority of the papers do not consider these objectives in their work. They emphasize the need for future research in this direction that operates within a robust ethical framework.

The topic is quite interesting and although the results might not be ground breaking at the moment, it deserves to be published in order to enhance discussion and research in that direction.

The paper is well written and easy to understand. The systems are well described. I have only some minor comments in order to enhance the flow of thoughts.

  • p. 3, r. 83: it would be interesting to mention the five industrial sectors
  • p. 3, r. 87: number of section is missing
  • p. 4, Fig. 1: I could not find a reference to the figure in the text
  • p. 8, r. 251: It would be great if you could provide an explanation/justification as to why the well-being of a person has a high impact on occupational safety.
  • p. 9, r. 304-311: It would be nice if you could shortly describe the intended use/purpose of the systems.
  • p. 14, r. 538: Could you please justify why the AI act is addressing only lawful AI but not ethical and robust AI?
  • p. 15, r. 612-614: Could you please elaborate a bit on why ensuring high data quality is not sufficient to ensure fairness in your context? Perhaps you could give some examples.
  • p. 16, r. 626: I wonder if the fact that you rely on the system to warn you of dangerous situations is not in itself a safety hazard if the system fails. Maybe you could add your thoughts on this?
  • p. 17, r. 660-663: It is not clear why you mention this under security and not under privacy. It would be helpful if you explained this a bit more.
  • p. 17, r. 665 and 666: Please double-check for typos

Finally, looking at the date of publication of the ISO and the AI act and that of the articles, one wonders if it is possible that the authors were aware of the concept and goals of trustworthy AI when they wrote their articles. Could it be that after the implementation of the AI act, authors and developers are more aware of these goals and are addressing them accordingly? What do you think?

Author Response

Thank you very much for your refined and helpful review!

Comment 1: “p. 3, r. 83: it would be interesting to mention the five industrial sectors“

Response 1: Thank you for this comment, yes of course. Those are oil and gas, mining, transportation, construction, and agriculture. I added them in the text.

Comment 2: "p. 3, r. 87: number of section is missing“

Response 2:  Thank you, fixed it in the text.

Comment 3: p. 4, “Fig. 1: I could not find a reference to the figure in the text“

Response 3: Thank you a lot for calling my attention to that mistake in referencing the figures, I corrected it.

Comment 4: “p. 8, r. 251: It would be great if you could provide an explanation/justification as to why the well-being of a person has a high impact on occupational safety.“

Response 4: Thank you for pointing this out. I added text on p.8 lines 251-255 to elaborate on the reason.

Comment 5: “p. 9, r. 304-311: It would be nice if you could shortly describe the intended use/purpose of the systems.”

Response 5: I elaborated the sentence and changed the text, adding a sentence about the use/purpose.

Comment 6: “p. 14, r. 538: Could you please justify why the AI act is addressing only lawful AI but not ethical and robust AI?”

Response 6: Thank you for this comment but I see its justification already described in the text. Since the AI Act of the European Union is the first legal regulation worldwide for AI, it is simultaneously the only one regulating AI from a legal viewpoint, which is a challenging and risky task (for that reason, there are no other law-binding regulations yet). All the other regulations are guidelines.

Comment 7: “p. 15, r. 612-614: Could you please elaborate a bit on why ensuring high data quality is not sufficient to ensure fairness in your context? Perhaps you could give some examples.”

Response 7: This is a good point, thank you. Besides data considerations which are of course very important, I added some examples of how a system design can ensure fairness.

Comment 8: “p. 16, r. 626: I wonder if the fact that you rely on the system to warn you of dangerous situations is not in itself a safety hazard if the system fails. Maybe you could add your thoughts on this?”

Response 8: This is a major and interesting point of discussion in our research group, however, this question goes under the category of accountability (who is responsible if the system fails) and robustness (is it likely that it fails under certain conditions and what to do about it). If the system fails and the user is overreliant on it, risks of different kinds might occur. However in our paper we are looking at assistance systems and not systems that are fully automated. When interacting with an assistance system, the responsibility lies with the human beings. With automated systems this question is still lergely unsolved globally. We chose to not elaborate on it in this paper since it would go beyond the scope, but it is, a challenging and relevant topic.

Comment 9: “p. 17, r. 660-663: It is not clear why you mention this under security and not under privacy. It would be helpful if you explained this a bit more.“

Response 9: Thank you, I have added a clarification on what is the main difference between security and privacy in the text.

Comment 10: “p. 17, r. 665 and 666: Please double-check for typos”

Response 10: Thank you, the typos has been found and fixed.

Comment 11: “Finally, looking at the date of publication of the ISO and the AI act and that of the articles, one wonders if it is possible that the authors were aware of the concept and goals of trustworthy AI when they wrote their articles. Could it be that after the implementation of the AI act, authors and developers are more aware of these goals and are addressing them accordingly? What do you think?“

Response 11: We are currently in the midst of a dynamic period where guidelines and regulations for AI are being developed, refined, and adapted to keep pace with rapid technological advancements. Some frameworks, such as the Ethics Guidelines for Trustworthy AI (2019) have been around at the publication times of the papers. Also, the preliminary drafts of the AI Act have already been around. Therefore, authors of papers could have made references to these or other guidelines.

It is true that the AI Act entered into force in 2024 and parts of it entered into force only in February 2025 and will enter into force in August 2026. We cannot know if researchers, when publishing their papers, paid attention to any of the guidelines, since research is mostly not abliged to follow any guidelines. For us as authors however, it was important to question wherther it makes sense to develop systems which will eventually have to comply with regulations, if compliance is not considered from the start? Integrating safety and ethical considerations early on is not only a proactive approach but also ensures smoother alignment with future legal requirements, ultimately fostering responsible and sustainable AI development. Therefore, we hope, this approach is going to change and safety will become a considerable part when researching AI systems and we hope we cast a vote for this with our paper.

Thank you again for taking your time. 

Reviewer 3 Report

Comments and Suggestions for Authors

This review aims to synthesize publications investigating AI-based safety systems (sensor-based and human-centered applications) at the research and development stage in order to assess how potential risks are addressed early in their design and prototype stages.

The authors stated two main research questions (Lines 173-175): RQ1: How can AI-based systems and technologies implemented for occupational safety in industrial settings be categorized? / RQ2: Given that some AI-based applications for occupational safety inherently introduce new risks while others do not, does the respective scientific literature address these risks?

Topic is very interesting and less presented in existing literature, with a lot of implication in practice of occupational safety in industrial domains (the AI based safety systems have an impact in identifying risks and supporting decision-making, creating safer work environments.

The methodology is clearly described in Chapter 2. Figure 1 and Figure 2 provide a clear picture of the methodological procedures used in the literature review process. Following a clear screening and selection process of the papers published after 2022 in Google Scholar, IEEE Xplore and ACM Digital Library databases, the authors finally reached a number of 20 relevant studies. The research method is complex and is well described. The authors also mention the limitations in research in chapter 4.2.

Questions: what was the reason behind selecting the Google Scholar, IEEE Xplore and ACM Digital Library databases and not WoS (Web of Science) and Scopus databases (these databases are very much used in academia).

The authors specifically answered the research objectives, de results being both described in detail and further on summarized. At the same time, the authors highlight the theoretical and practical implications.

The authors are using relevant, updated literature references, pertaining to the investigated topic.

Author Response

Thank you for taking the time to review our paper and for highlighting its positive aspects. We truly appreciate your thoughtful feedback and recognition of our work.

As to your question why we selected Google Scholar, IEEE Xplore and ACM Digital Library databases and not WoS or Scopus: there is no particular reason reason behind it other than me being the main author and a young researcher, at the starting point of writing and researching for this paper, have been acqainted with the chosen databases since they were used the most in our computer science based research community and thus they were suggested to me.